ə | **Open Peer Review** | Parasitology | Research Article

# A laboratory-adapted and a clinical isolate of dengue virus serotype 4 differently impact *Aedes aegypti* life-history traits relevant to vectorial capacity

Mariana Maraschin,[1] Diego Novak,[1] Valdorion José Klein Junior,[1] Lucilene W. Granella,[1] Luiza J. Hubner,[1] Athina R. Medeiros,[1] Tiago Gräf,[2] Guilherme Toledo-Silva,[3] Daniel S. Mansur,[1] José Henrique M. Oliveira[1,4,5]

**ABSTRACT**  Dengue virus cases are on the rise globally, and strategies to control its primary vector, the mosquito *Aedes aegypti*, represent a promising approach. However, the interaction between virus serotypes and genotypes with *Ae. aegypti* is poorly characterized in terms of life-history traits related to vector capacity and mosquito disease tolerance. Here, we infected *Ae. aegypti* with two phylogenetically distant strains of dengue virus serotype 4 genotype II: a laboratory-adapted strain, DENV4-TVP/360, and a recent clinical strain isolated from southern Brazil, DENV4-LRV 13/422. These strains exhibit 26 amino acid differences in their sequences. We assessed various life-history traits of *Ae. aegypti*, including mortality, fecundity, fertility, and induced flight capacity, as well as vector competence-related parameters, such as infection intensity and prevalence, following exposure to different viral concentrations. We found that while neither strain significantly reduced mosquito lifespan, infection prevalence was highly influenced by the initial dose of DENV4-TVP/360. In contrast, the infectious prevalence of DENV4-LRV 13/422 was smaller (~40% at 14 days post-infection [DPI]), regardless of the initial virus titers. The DENV4-TVP/360 strain also enhanced mosquito-induced flight capacity at early (1 day post-infection) and late (21 DPI) time points, when dispersion is critical for vector competence. On the other hand, DENV4-LRV13/422 reduced *Ae. aegypti* fertility. A better understanding of how arbovirus strains influence mosquito life-history traits connected to disease spread is critical in public health efforts to mitigate arbovirus outbreaks that are focused on the mosquito vector.

**IMPORTANCE**  As dengue virus (DENV) cases rise globally, vector control strategies targeting *Aedes aegypti* remain essential to public health. The effectiveness of these interventions depends on understanding how viral strains interact with mosquito biology. We show that two phylogenetically distinct DENV serotype 4 strains—one laboratory-adapted, one recently isolated—differentially affect *Ae. aegypti* traits, such as fertility, flight capacity, and infection dynamics. These traits are linked to mosquito fitness and vector competence, and our results show that sequence variation can shape mosquito-virus interactions. The lab strain enhanced flight capacity at key time points, potentially aiding transmission, while the clinical isolate reduced fertility independently of viral dose. Infection prevalence and sensitivity to viral dose also differed between strains. These findings highlight how viral genotype influences mosquito performance and transmission potential. Incorporating viral genetic diversity into arbovirus studies can improve disease spread models and inform mosquito-based control strategies.

**KEYWORDS**  *Aedes aegypti*, life-history traits, dengue virus, vector competence

**Peer Reviewer** Wei Huang, Chinese Academy of Sciences Shanghai Institute of Immunity and Infection, Shanghai, Shanghai, China

Address correspondence to José Henrique M. Oliveira, jose.oliveira@ufsc.br.

Mariana Maraschin and Diego Novak contributed equally to this article. The order of authors was determined based on their contributions to the article.

The authors declare no conflict of interest.

See the funding table on p. 11.

The ability of hosts to sustain infection without significant impact on their health and fitness is called disease tolerance, a key feature of vector competence and pathogen transmission by mosquitoes (1). *Aedes aegypti* is predominantly tolerant to the arboviruses they transmit, such as dengue and Zika (2, 3). Several life-history traits of mosquitoes, such as survival, reproductive output, and dispersal, influence arbovirus transmission (4) and can be used to quantify mosquito disease tolerance (5). Well-defined mosquito health metrics are critical for a better understanding of the molecular mechanisms that enable *Ae. aegypti* to sustain a chronic arbovirus infection during its life cycle. Identifying and characterizing such mechanisms and how their manipulation changes transmission is fundamental to developing strategies to block vector competence and arbovirus epidemics (6, 7).

We evaluated the impact of two dengue virus serotype 4 genotype II strains (DENV4) on different life-history traits of *Ae. aegypti*. The strains are a laboratory-adapted reference strain DENV4-TVP/360 (GenBank: KU513442), originally isolated in the Dominican Republic in 1981 (8, 9) and a clinical isolate from a patient presenting high serum viral load in southern Brazil, DENV4-LRV 13/422 (GenBank: KU513441) (10, 11). They were chosen because they exhibit different *in vitro* infectivity rates and immuno-modulatory capacities in vertebrate cells (11). Specifically, the LRV 13/422 strain is able to infect C6/36 *Ae. albopictus* cells and Huh7.5 hepatocyte-derived human cells more efficiently than the TVP/360 strain. Also, LRV 13/422 is less sensitive to interferon-alpha (IFN-α) inhibition in Huh7.5 cells, and preliminary analyses revealed an extensive genetic divergence between the two strains (11).

Based on differences in *in vitro* infection efficiency and immune responses in vertebrate cells, we hypothesized that the two strains would differentially affect *Ae. aegypti* life-history traits under laboratory conditions. Initially, we analyzed in more detail the virus genome of both strains, finding that they were phylogenetically distant with 26 differences in their amino acid sequences. Then, we measured *Ae. aegypti* life-history traits after infection with the DENV strains, focusing on parameters that are directly linked to mosquito fitness, such as survival, fecundity, and induced-flight activity (INFLATE), a proxy for dispersion (12). We found strain-specific phenotypes in the infection prevalence, fertility, and INFLATE at certain time points following infection. To modulate mosquito disease tolerance, we need to improve our ability to quantify mosquito fitness and homeostatic status during infection and to understand the molecular basis of strain-specific effects on vector life-history traits (13–15).

## MATERIALS AND METHODS

### Mosquitoes

*Ae. aegypti* (Red Eye strain) were reared and maintained under standard conditions as described previously (16, 17) at the Federal University of Santa Catarina, Brazil, with a 12 h light/dark cycle at 28°C and 70%–80% relative humidity. Larvae were maintained in filtered, dechlorinated tap water at a density of approximately 100 larvae per liter. They were fed Purina powdered dog chow, with a total of 500 mg provided in 2–3 additions over the course of the larval stage, without changing the water. Adults were provided a 10% sucrose solution *ad libitum* while housed in 7.3 L plastic cages (21 cm diameter × 25 cm height) with ~200–300 mosquitoes per cage. Infections were performed in females aged between 3 and 7 days post-adult eclosion, which were used in all assays and maintained under the same light-dark cycle at 28°C and 80% relative humidity. Additionally, for egg production to maintain the insectarium, females were fed artificially using a water-jacketed glass artificial feeder and a parafilm membrane containing peripheral human blood (collected into 5 mL tubes containing 7 mg of $K_2$-ethylenediaminetetraacetic acid (EDTA) and 1 mM adenosine triphosphate (ATP) (pH 7) as a phagostimulant. Informed consent was obtained from all blood donors. This protocol was approved by the Federal University of Santa Catarina (UFSC)—CAAE: 89894417.8.0000.0121.

## Dengue virus serotype 4 stock preparations

We used dengue virus 4 strain TVP/360—GenBank: KU513442, hereafter called DENV4/TVP, and dengue virus 4 strain LRV13/422—GenBank: KU513441 at P5 with a known history passage, hereafter called DENV4/LRV (11). Virus strains were kindly provided by Dr. Claudia Nunes Duarte dos Santos from Instituto Carlos Cagas—Fiocruz Paraná, Brazil. *Ae. albopictus* mosquito cells of the C6/36 lineage (ATCC, CRL-1660) were maintained and propagated in 1× L-15 medium with a pH of 7.6, supplemented with 5% SFB, 1% P/S, and 0.26% tryptone, at a temperature of 28°C in a Biochemical Oxygen Demand (B.O.D.) incubator. The cells were seeded at 80% confluency ($3 \times 10^7$ cells) in a 150 cm$^2$ bottle, which was kept at 28°C overnight. The following day, the cells were infected with a multiplicity of infection of 0.01 for 5 days to produce DENV4/TVP or DENV4/LRV. The cell supernatant, containing viral particles, was collected, centrifuged at $460 \times g$ for 10 min at 4°C, aliquoted, and stored at −80°C. The plaque assay was performed to determine the viral titer.

## Phylogenetic analysis

A comparative phylogenetic analysis between LRV 13/422 and TVP/360 strains was performed by using augur/auspice as available in Nextstrain (18). Initially, all DENV4 genomes with more than 70% coverage compared to the reference strain (NC_002640) were retrieved from NCBI Virus. Sequences were then aligned with MAFFT (19) and visually inspected in Aliview (20). A maximum likelihood phylogenetic tree was constructed with IQ-TREE (21) and the best substitution model was inferred by the model testing function. Branch support was calculated with SH-aLRT in 1,000 pseudoreplicates. The phylogenetic tree was then visualized in FigTree (https://github.com/rambaut/fig-tree), revealing that both LRV 13/422 and TVP360 strains and the DENV4 reference genome belonged to genotype 2. We then reconstructed the ancestral sequences of this genotype and translated the mutations using augur. The sequence AF326573 was used as the root sequence since NC_002640 was derived from an engineered strain (AF326825), and AF326573 was the natural isolation from which AF326825 originated (9). Auspice was used to visualize and stylize the tree and to extract the LRV 13/422 and TVP/360 mutation path from the root sequence.

## *Ae. aegypti* infection with dengue virus serotype 4

For mosquito infection experiments, human blood was collected from healthy donors (UFSC-CAAE: 89894417.8.0000.0121) using tubes containing EDTA reagent. To isolate red blood cells (RBCs), 1 mL of blood was centrifuged at 6,400 rpm for 4 min at room temperature. Following centrifugation, the serum was discarded, and the cells were gently washed twice with sterile 1× phosphate-buffered saline solution (GIBCO). *Ae. aegypti* females were fasted for 18–24 h before a blood meal. During this fasting period, mosquitoes had access to water (but not sucrose solution) before being offered a meal containing a 1:1 mixture of RBC cells and L-15 medium with either DENV4 TVP/360 virus or DENV4 LRV13/422. The control group (Mock) received a 1:1 mixture of RBC and C6/36 cell supernatant. All groups included ATP, pH 7.4, at a final concentration of 1 mM as a phagostimulant. These solutions were presented to the females using artificial glass feeders heated by water at 37°C for approximately 1 h. Subsequently, mosquitoes were anesthetized using cold (−20°C), and only fully engorged females were separated into small cages and kept in a B.O.D. incubator at 28°C until the experiment concluded. Each cage housed 20 mosquitoes and was supplied with a 10% sucrose solution soaked in cotton, offered *ad libitum*.

## Plaque assays

Vero cells (ATCC, CCL-81) were used to quantify DENV4 TVP/360 and Vero E6 cells (ATCC, CRL-1586) were used to quantify DENV4 LRV 13/422 based on plaque optimization for each viral strain. Cells were seeded at a density of $5 \times 10^4$ cells/well and $1 \times 10^5$ cells/

well, respectively, in a 24-well plate in Dulbecco's Modified Eagle Medium (DMEM) F-12 supplemented with 5% FBS, 1% penicillin/streptomycin, 1% glutamine (1× complete DMEM) maintained at 37°C and 5% $CO_2$. The following day, the mosquito samples were thawed and underwent a decontamination process, which involved immersing each mosquito in $1 \times 45''$ in 70% alcohol, followed by $1 \times 45''$ in 1% hypochlorite, another round of $1 \times 45''$ in 70% alcohol, and finally $1 \times 45''$ in 0.9% sterile saline. Subsequently, each mosquito was transferred to a 1.5 mL tube containing 200 µL of complete DMEM F-12 medium (as defined above) and kept on ice. The mosquitoes were then macerated individually using a manual vortex with a sterile pestle dedicated to each sample. The homogenate was then centrifuged at $3,200 \times g$ for 5 min at 4°C. Each mosquito homogenate was diluted (ranging from $10^{-1}$ to $10^{-5}$) in DMEM F-12-1× complete medium, added to Vero cells, and incubated in 200 µL of each dilution for 60 min at 37°C and 5% $CO_2$. The same procedure was performed for the control group (Mock) using 200 µL of complete DMEM F-12 medium. After incubation, the medium containing the viral dilutions was removed, and 700 µL of semi-solid medium containing DMEM 2× supplemented with 1% fetal bovine serum, 1% P/S, and 1.6% carboxymethyl-cellulose was added. The plates were then incubated for 5 days at 37°C and 5% $CO_2$. Following incubation, the cells were fixed in paraformaldehyde 3% for 20 min, stained in 1% crystal violet, and counted.

## Survival curves

All infections were performed in female mosquitoes between 4 and 6 days following adult emergence. Infected females were cold-anesthetized immediately after feeding and transferred to cardboard cups with a density of 20 fully engorged females per cup (maximum capacity of 470 mL—10 cm height × 9 cm diameter). *Ad libitum* access to a 10% sucrose solution was provided through cotton pads placed on top of a woven mesh, which were replaced 2–3 times weekly. Survival rates were monitored daily, six times a week, until all insects within the cups died. The survival cages were maintained in the insectary at a temperature of 28°C (±10%) and humidity of 80% (±10%). Survival data presented represent pooled results from a minimum of two independent experiments.

## *Ae. aegypti* fecundity and fertility

Female mosquitoes were artificially fed with whole blood, mock, or blood supplemented with DENV4-TPV/360 or DENV4-LRV 13/422 as detailed in section "*Ae. aegypti* infection with dengue virus serotype 4." Fully engorged females were cold-anesthetized and individually housed in cages containing a dark plastic cup with water and filter paper to allow egg deposition. Fecundity was scored 5 days following the meal by counting the number of eggs deposited per female. Collected eggs were allowed to rest in insectary conditions for 7 days, when they were submerged in water containing filtered and dechlorinated tap water plus powdered dog chow to allow larval development. Fertility was assessed 3 days post-eclosion by counting the percentage of eclosed larvae.

## *Ae. aegypti* induced-flight activity

The protocol was adapted from reference 12. A rectangular plastic cage measuring 19 cm × 20 cm was divided into four equal quadrants, in addition to the base. Each quadrant was assigned a value ranging from zero, the lowest (base), to four, the highest. Five mosquitoes were quickly anesthetized at a temperature of −20°C and introduced into the cage, where they remained for 10 min, acclimating to the experimental conditions in the insectary temperature and humidity. To begin the INFLATE test, a mechanical stimulus was applied to the cage, consisting of lifting it about 20 cm above the surface and gently tapping it, causing the mosquitoes to detach and fall to the base of the cage. The cage was kept stable for 1 min, during which time the mosquitoes initiated flight activity. The number of mosquitoes that landed in each quadrant of the cage was visually recorded. Mosquitoes still flying after 1 min were assigned the highest value

(four). Mosquitoes landing on the line between quadrants received the value of the upper quadrant. Then, a 2 min rest period was observed, and the process was repeated. Each repetition generated a value called the "inflate value," calculated by multiplying the value of each quadrant by the number of mosquitoes landing in it. These values were then summed and divided by the total number of mosquitoes in the cage (five). This process was repeated 10 times, corresponding to a technical replicate ($n = 10$). The average of the values from the 10 technical replicates is referred to as the "Inflate Index." The protocol was conducted with infected and non-infected mosquitoes (control group, mock). Ultimately, a biological replicate was obtained ($n = 1$). In Fig. 4, each dot represents one biological replicate (consisting of five mosquitoes assayed 10 times).

## Statistical analysis

Infection intensity, day of death, eggs per female, percentage of eclosion, and INFLATE index were tested for normality. For data that did not follow a normal distribution, non-parametric tests, such as the Kruskal-Wallis test, were performed, as indicated in the figure legends, for side-by-side comparisons or Dunn's multiple comparisons test. Infection prevalence was analyzed using a contingency table (Yes—infected vs No—uninfected) and Fisher's exact test. Survival curves were analyzed with a Log-rank (Mantel-Cox) test. Statistical analysis was conducted using GraphPad Prism.

## RESULTS

### Sequence differences between DENV4-TVP/360 and DENV4-LRV 13/422 sequences

We compared the sequences of the laboratory-adapted (DENV4-TVP/360) and the clinical isolate (DENV4-LRV 13/422) strains of DENV4 with a reference strain, DENV4/814669, obtained from the Dominican Republic in 1981 (8, 9). Both strains were phylogenetically distant (Fig. S2) with the clinical isolate (LRV 13/422) exhibiting 24 unique amino acid (AA) substitutions compared to the reference sequence DENV4/814669, while the laboratory-adapted (TVP/360) strain exhibited two unique AA substitutions, compared to DENV4/814669 (Fig. S1; Table S1). Three mutations were shared by both strains: S2H, in the precursor membrane glycoprotein (prM), K14Q, in the non-structural protein 3 (NS3), and R23K in the non-structural protein 5 (NS5). The two exclusive amino acid substitutions of the DENV4-TVP/360 strain were located in the non-structural protein NS4B, being phylogenetically closer to the reference strain, DENV4/814669, than the virus isolated from southern Brazil in 2013 (DENV4-LRV 13/422). Most of the non-synonymous mutations of the Brazilian strain (LRV 13/422) were present in the envelope protein (E), with six AA substitutions; non-structural protein 1 (NS1) with five AA substitutions; non-structural protein 2A (NS2A), with five AA substitutions, and the non-structural protein 5 (NS5), with six AA substitutions. Interestingly, these proteins are critical for flavivirus infection, replication, and dissemination into mosquito tissues (22–27).

### Infection dynamics of DENV4 strains differ in *Ae. aegypti*

To explore how the laboratory-adapted (DENV4-TVP/360) and the clinical isolate (DENV4-LRV 13/422) interact with *Ae. aegypti* over time, we infected mosquitoes with two doses of each strain and measured virus titers in single whole-body mosquitoes at 0, 7, 14, and 21 days following the infectious blood meal (Fig. 1A and B). Since the initial infectious dose strongly influences DENV infection in *Ae. aegypti* (28), we used the maximum available dose for each strain and a 1/10 dilution. The infectious particle virus concentration (input in Fig. 1C) is shown in red and expressed as plaque-forming units (PFU) per microliter of blood offered to mosquitoes (PFU/µL of blood). Both strains were able to infect and replicate in *Ae. aegypti* (Fig. 1B). Using the same infectious dose when comparing infection dynamics between groups is essential, and our experimental design accounted for this. There is no statistical difference in the viral titers in the blood meals provided to mosquitoes between the DENV4 TVP "low dose" and DENV4 LRV

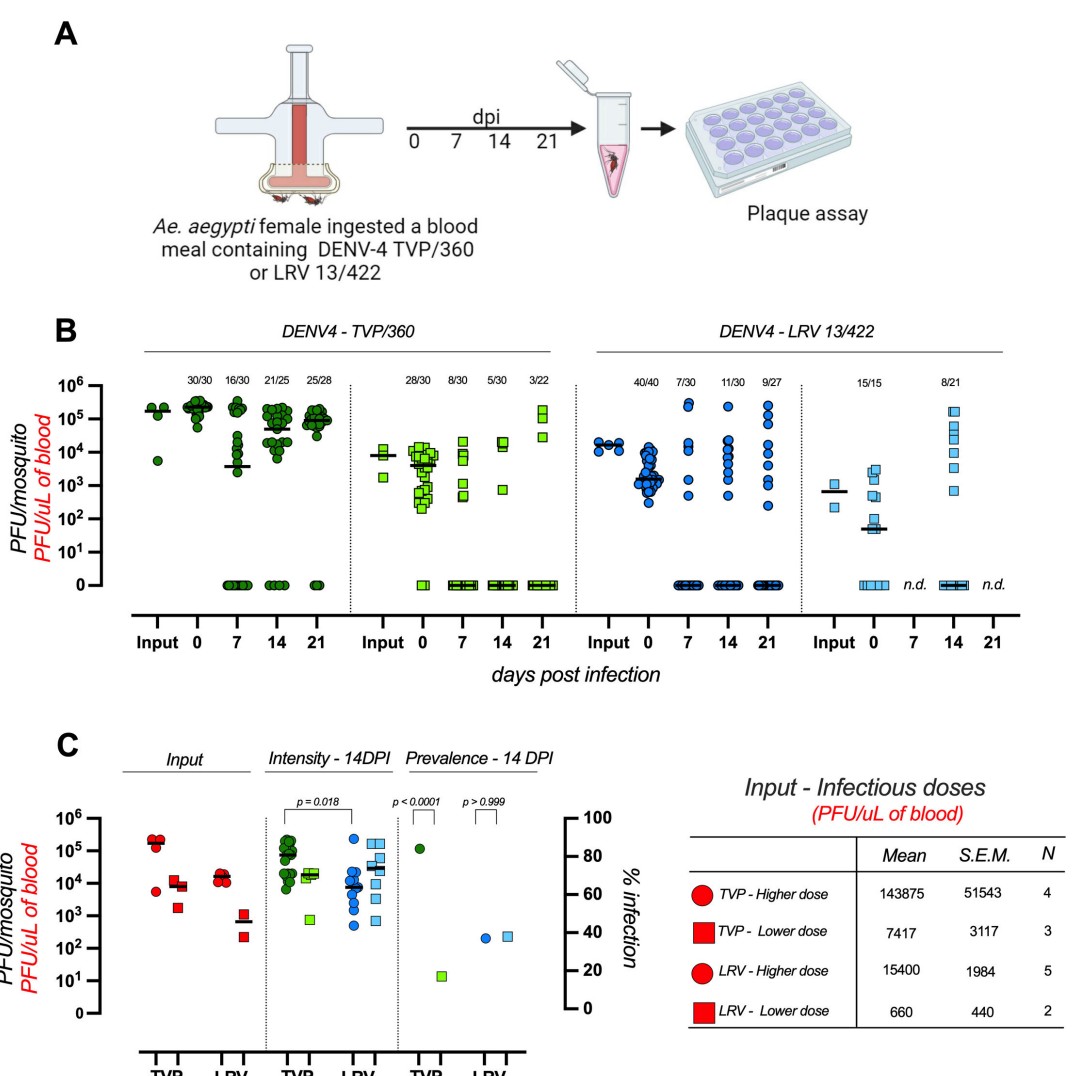

**FIG 1** DENV4 TVP/360 and LRV 13/422 infection dynamics in *Ae. aegypti* mosquitoes. (A) Experimental design. Mosquitoes were fed with human blood containing different concentrations of two strains of DENV4-TVP/360 and DENV4-LRV 13/422. (B) Viral load was measured by plaque assay immediately following feeding (0), 7, 14, and 21 days post-infection (DPI). Each point represents one female mosquito. The number of samples tested is indicated above each group. Red dots represent the virus concentration in the blood offered to mosquitoes in each replicate experiment. Bars in each column represent medians. This figure represents the sum of three to four independent experiments for each group. N.d.: not determined. (C) Input is defined as the infectious particle concentration presented in the blood meal offered to mosquitoes (expressed as PFU/µL of blood—red text in the left Y-axis). Intensity is defined as the number of infectious particles per mosquito measured at 14 days post-infection (PFU/mosquito in the left Y-axis). Prevalence is defined as the percentage of infected mosquitoes at 14 days post-infection (% infection in the right Y-axis). Statistical analysis: intensity—Kruskal-Wallis test. Prevalence—Fischer's exact test.

"high dose" groups, as shown by the analysis of independent biological replicates (Fig. S3A). Consistently, there is also no statistical difference in the amount of infectious virus ingested by *Ae. aegypti* immediately after feeding (Fig. S3B).

We measured two readouts: infection prevalence, the percentage of infected mosquitoes, and infection intensity, the amount of infectious particles per mosquito. Overall, infection intensity showed no dose-dependent increase with input dose in all the experimental conditions. For the DENV4 TVP strain, mosquitoes ingested either $2.2 \times 10^5$ PFUs ($\pm 1.3 \times 10^4$ standard error of the mean [S.E.M.], $n = 30$) at the highest dose or $4.8 \times 10^3$ PFUs ($\pm 8.3 \times 10^2$ S.E.M., $n = 28$) at the lowest dose. Despite this 46,000-fold difference in input, infection intensity at 14 DPI was not significantly different ($P = 0.326$, K–S test). The same pattern was observed with the DENV4 LRV strain. At the highest dose, mosquitoes ingested $3.6 \times 10^3$ PFUs ($\pm 6.0 \times 10^2$ S.E.M., $n = 40$), compared to $5.4 \times 10^2$

PFUs ($\pm 2.5 \times 10^2$ S.E.M., $n = 15$) at the lowest dose. Here too, despite a 6.6-fold increase in virus input, infection intensity at 14 DPI did not differ ($P > 0.999$, K–S test). Taken together, input doses ranging across ~$4 \times 10^5$ PFUs (a 407,781-fold difference between the TVP high dose and the LRV low dose) had little effect on virus titer per mosquito at 14 DPI. The only significant difference was detected at 14 DPI when comparing the highest doses of TVP and LRV. However, this range of input doses differentially influenced infection prevalence depending on the virus strain, with the laboratory-adapted strain (DENV4-TVP/360) being highly influenced by virus input, as expected based on several mosquito-DENV studies (28–30). However, the clinical isolate (DENV4-LRV 13/422) exhibited a relatively low infection prevalence (~40%), irrespective of the input dose (Fig. 1C). This result suggests that factors determining infection prevalence, also known as the midgut infection barrier, might be more relevant to vector competence than immune resistance factors that decrease virus replication inside mosquito tissues once the infection is already established (31–33).

## The impact of different DENV4 strains on life-history traits of *Ae. aegypti*

We challenged mosquitoes with two doses of DENV4/TVP and DENV4/LRV and scored mosquito survival during the full life span of the population. Consistent with our previous results (5), the median time to death of *Ae. aegypti* did not differ between uninfected (mock) and infected mosquitoes, regardless of input viral doses and virus strain (Fig. 2).

The impact of virus infection on parameters directly connected to vector fitness was evaluated following mosquito challenge with both DENV4 strains. Feeding *Ae. aegypti* with blood supplemented with different concentrations of infectious particles did not alter the total number of eggs laid by each fully engorged female (Fig. 3A). In Fig. 3, "blood" represents whole human blood (RBCs + plasma), which has a higher total protein concentration than mock, as evidenced by a greater fecundity of *Ae. aegypti* (blood ~70 eggs/female vs mock ~35 eggs/female). Challenging mosquitoes with the clinical strain of DENV4 (LRV) resulted in a significant reduction in the percentage of eclosion (~50%), irrespective of the dose tested (Fig. 3B). This result suggests a lower adaptation of the LRV strain, which is consistent with its recent interaction with our colony mosquitoes (Red Eyes strain) instead of field mosquitoes circulating in southern Brazil (34). On the other hand, we did not observe statistically significant differences in fertility with the laboratory-adapted TVP strain, consistent with the higher adaptability of this virus under laboratory conditions. Interestingly, there is no difference in fertility between blood-fed and mock-fed mosquitoes, averaging around 70% eclosion, suggesting that females can optimize fecundity (egg production) according to the nutritional status of the meal to maximize fertility, similar to what has been described for desiccation stress in *Ae. aegypti* (35) .

*Ae. aegypti* dispersal involves its flight activity and is directly connected to vectorial capacity and arbovirus epidemics (36). We took advantage of a recently described method to assess the induced flight activity (INFLATE) (12) and challenged *Ae. aegypti* females with the highest available doses of both strains of DENV4. The laboratory-adapted TVP/360 strain significantly enhanced the mosquito flight activity 24 h after feeding when compared to "mock" and DENV4-LRV 13/422 (Fig. 4B), a time point where infection of the midgut epithelium is taking place. At 21 days post-feeding, an epidemiologically relevant time point, where DENV4 has already infected the salivary glands and the mosquito is infectious (37, 38), the INFLATE index was also significantly higher, specifically in the laboratory-adapted TVP strain (Fig. 4E). The clinical isolate did not induce alterations in the flight activity during the mosquito lifespan, compared to mock-infected *Ae. aegypti* (Fig. 4B through E). We observed an overall decline in the INFLATE index 1 day post-feeding (Fig. 4B) as reported by reference 12, compared to all the other time points. This reduction was independent of the DENV4 challenge. After the completion of blood digestion, at days 7, 14, and 21 in our assays, mosquitoes were leaner and lighter, which translated into higher INFLATE values compared to 1 DPI (Fig. 4C through E). At 21 days

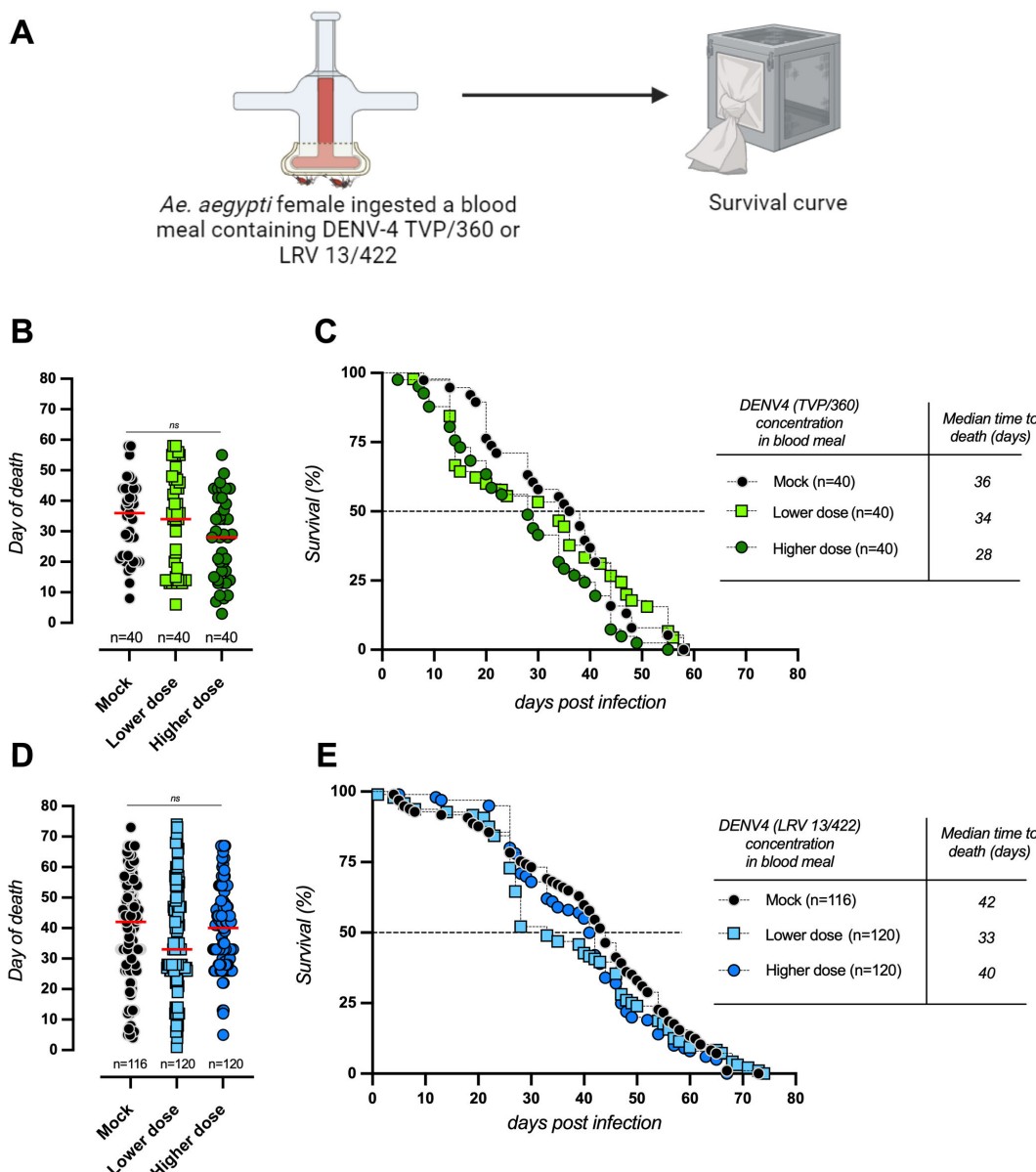

**FIG 2** Survival curves of *Ae. aegypti* infected with DENV4 strain TVP/360 and DENV4 strain LRV 13/422. (A) Experimental scheme. Three to four days following adult emergence, females were fed with blood supplemented with two concentrations of DENV4 strains, as indicated in panel C (input—infectious doses—PFU/µL of blood). (B through E) Survival curves were performed at least twice in batches of 20 fully engorged females per cage. Survival was scored six times per week until all the mosquitoes died. Note that the survival of DENV4 (TVP/360) was already tested in much higher numbers in our previous work with similar results (5). Horizontal bars in panels B and D represent medians. Statistical analysis in Fig. 3B and D—Kruskal-Wallis test.

post-feeding, *Ae. aegypti* is already experiencing senescence, causing a reduction in the overall INFLATE index compared to 7 and 14 DPI (Fig. 4E).

## DISCUSSION

Mosquitoes can harbor and transmit microbes, such as dengue virus and Plasmodium spp., without exhibiting overt signs of pathology, a phenomenon often described as disease tolerance. While this trait has been relatively well characterized phenotypically, its underlying molecular mechanisms remain poorly understood (1, 2, 39, 40). In our previous work, we explored the impact of different arboviruses on mosquito mortality

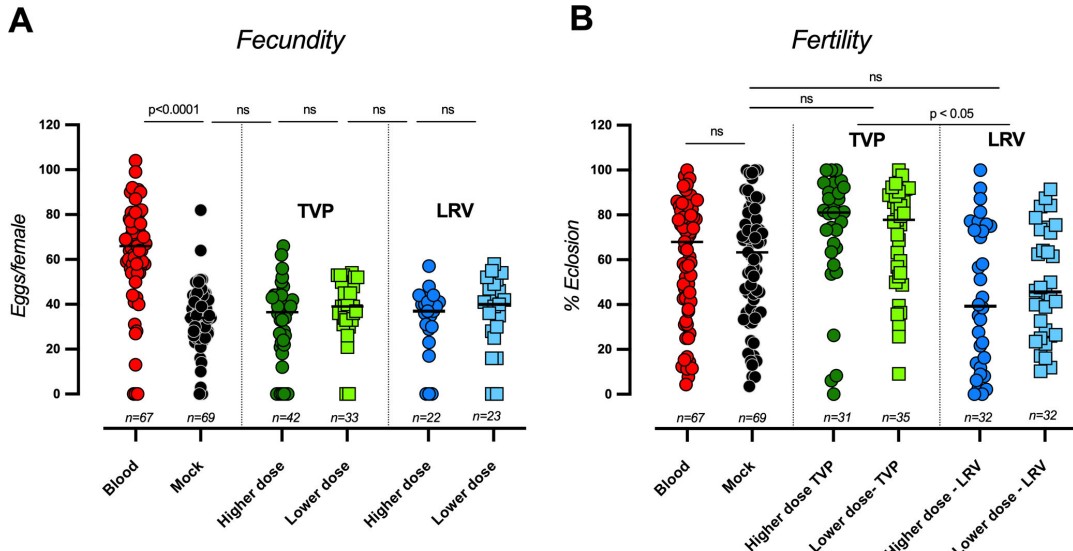

**FIG 3** Fecundity and fertility of *Ae. aegypti* mosquitoes infected with DENV4 strains TVP/360 and LRV 13/422. Female mosquitoes were fed with human blood containing two concentrations of DENV4 strains, TVP or LRV, as indicated in Fig. 2C (input—infectious doses—PFU/µL of blood). (A) Each point represents fecundity (eggs/female) measured through the individual oviposition of each mosquito. (B) Fertility represents the percentage of viable eggs laid by each mosquito individually measured 7 days post-egg laying. Mosquitoes infected with DENV4 were compared with their respective controls: blood and mock (RBCs containing C6/36 cell supernatant instead of serum, as detailed in the Materials and Methods). The number of samples tested is indicated at the bottom of each column. Experiments were conducted at least three times and presented mean plus standard error. Statistical significance was determined using the Kruskal-Wallis test, followed by Dunn's multiple comparison test.

to establish a dose-response framework to study *Ae. aegypti's* response to infections (5). A quantitative analysis of disease tolerance can be achieved by scoring how parameters relevant to host physiology vary during a gradient of infection (14, 15, 41). In this manuscript, we investigated the effect of two distinct DENV4 genotype II strains on mosquito life-history traits useful to describe organismal fitness and disease tolerance. We found that both DENV4 strains differentially impacted infection prevalence, female fertility, and induced flight capacity. Defining the molecular basis that underlines such phenotypes will be critical in efforts to mitigate mosquito-borne diseases based on vector control.

Dengue virus serotype 4 is one of the least studied dengue viruses. Although its presence in Brazil has been relatively limited (42–45), studies on the dynamics of different serotype circulation and emergence (46), as well as the occurrence of antibody-dependent enhancement of disease in humans (47–49), highlight the importance of examining the interaction between DENV4 and its main vector, the mosquito *Ae. aegypti*. This is particularly relevant since DENV4 has been shown to circulate undetected in mosquito populations without reported human cases (50, 51), be vertically transmitted by mosquitoes (52), and displace DENV1, a common DENV serotype in Brazil (53) when co-infecting *Ae. aegypti* (54).

*Ae. aegypti* dispersal is a critical parameter to arbovirus infection in humans (55, 56). We found that DENV4-TVP/360, a laboratory-adapted strain, enhanced *Ae. aegypti* induced-flight capacity (INFLATE), a proxy for mosquito dispersal, specifically at an early (1 DPI) and late (21 DPI) time points (Fig. 4B through E). This modulation was specific to the TVP strain since the recent clinical isolate (LRV) did not show differences in the INFLATE index compared to mock-fed mosquitoes (Fig. 4B through E). It was previously shown that DENV infection enhances mosquito behaviors linked to flight (57–59). Recently, immune activation by zymosan was shown to reduce *Ae. aegypti's* induced flight capacity (60). The molecular and physiological basis of immune-induced flight phenotypes in *Ae. aegypti*, as well as its epidemiological implications, remains to be defined (61).

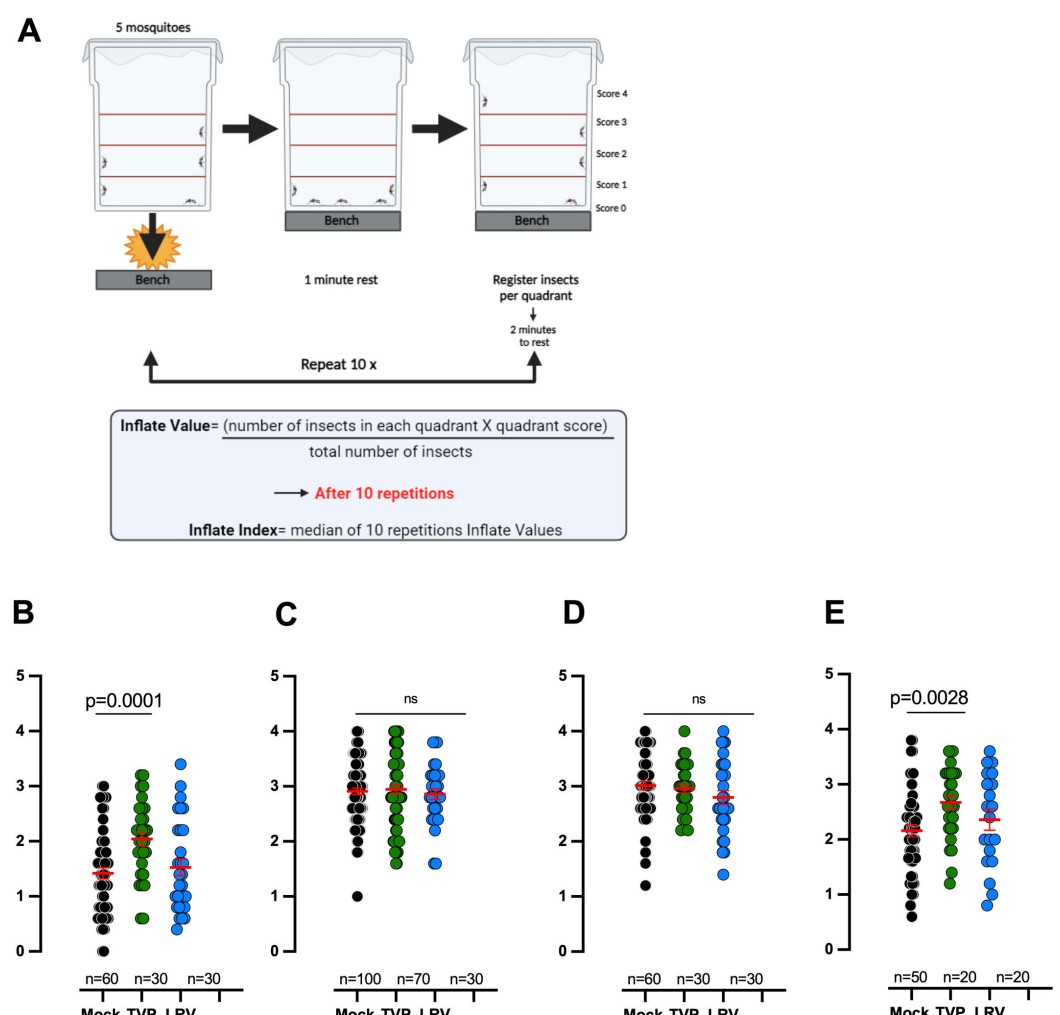

**FIG 4** Comparative analysis of the induced flight activity of *Ae. aegypti* mosquitoes infected with DENV4 strains TVP/360 and LRV 13/422. (A) Experimental design of the INFLATE assay. See details in the Materials and Methods. Female mosquitoes were fed on blood containing the highest doses available for each strain—see Fig. 2 (input). (B through E) The induced flight activity of each group was measured at 1, 7, 14, and 21 DPI. Mosquitoes challenged with DENV4 were compared with the mock/uninfected group. Each dot represents the inflate index (five mosquitoes per cage tested 10 times in a row). The experiments were conducted at least three times, and statistical significance was determined by the Tukey multiple comparison test and ANOVA.

The contribution of different defense strategies, such as antiviral resistance or disease tolerance, during mosquito immune response to DENV is poorly defined (1). While DENV load increases 100–1,000-fold in susceptible mosquito strains during its spread from the midgut to the salivary glands (37), *Ae. aegypti* does not experience significant fitness costs associated with chronic infection (5), although this may vary depending on different DENV serotypes, genotypes, and mosquito populations (62). The RNAi machinery is considered the main antiviral pathway in *Ae. aegypti* (63), but the knockout or overexpression of its core gene, Dicer-2, revealed conflicting results regarding its actual contribution to mosquito immune resistance to arbovirus infection and vector competence (64–66). In Fig. 1C, we show that despite a marked difference in prevalence at 14 DPI between the LRV 13/422 and TVP/360 strain, depending on the initial DENV4 input dose, infection intensity (Fig. 1C—middle panel) exhibited a modest dose-dependency. This result suggests that immune resistance plays a minor role in the DENV4 replication cycle within the mosquito at the whole-body level, and unknown factors

controlling the midgut infection barrier might be more important to mosquito infection and, therefore, disease transmission.

In summary, we described that two different strains of DENV4 genotype II differentially impacted *Ae. aegypti* infection dynamics, fertility, and flight capacity. The determination of how arbovirus infection affects life-history traits of insect vectors is critical for the development of strategies to fight mosquito-borne diseases (67). Some of the limitations of our study include the fact that we have demonstrated viral strain-induced changes in parameters associated with *Ae. aegypti* vector competence and physiology under laboratory conditions, and not in field settings, with the influence of real-world ecological parameters. The DENV4 TVP/360 strain is laboratory-adapted and reaches high titers in cell culture. It remains possible that the observed infection prevalence (Fig. 1C) and induced-flight activity (Fig. 4) result from artificial adaptation in cell culture. Also, feeding mosquitoes with decreasing doses of arboviruses leads to reduced percentages of infections (Fig. 1C). The observed differences should be interpreted with caution and understood as group or population effects.

## ACKNOWLEDGMENTS

This work was supported by Instituto Serrapilheira (#13452 to J.H.M.O.); CNPq (Conselho Nacional de Desenvolvimento Científico e Tecnológico) (#407312/2018 and #403499/2021-6 to J.H.M.O.) and CNPq/INCT-EM (Conselho Nacional de Desenvolvimento Científico e Tecnológico/Instituto Nacional de Ciência e Tecnologia em Entomologia Molecular). Fundação de Amparo à Pesquisa e Inovação do Estado de Santa Catarina (FAPESC) (#2024TR002383 to J.H.M.O. and 2024TR002667 to J.H.M.O.).

M.M.: Investigation; Data curation; Writing—review and editing. D.N.: Investigation; Data curation; Writing—review and editing. V.J.K.J.: Investigation; Data curation; Writing—review and editing. L.W.G.: Investigation; Data curation; Writing—review and editing. L.J.H.: Investigation; Data curation; Writing—review and editing. A.R.M.: Investigation. T.G.: Data curation; Methodology; Writing—review and editing. G.T.-S.: Data curation; Methodology. D.S.M.: Conceptualization; Data curation; Funding acquisition; Writing—review and editing. J.H.M.O.: Conceptualization; Data curation; Formal analysis; Funding acquisition; Project administration; Supervision; Writing—original draft.

## AUTHOR AFFILIATIONS

[1]Departamento de Microbiologia, Imunologia e Parasitologia, Universidade Federal de Santa Catarina, Florianópolis, Brazil

[2]Laboratório de Virologia Molecular, Instituto Carlos Chagas, Fundação Oswaldo Cruz, Curitiba, Brazil

[3]Departamento de Biologia Celular, Embriologia e Genética, Universidade Federal de Santa Catarina, Florianópolis, Brazil

[4]Instituto Nacional de Ciência e Tecnologia em Entomologia Molecular, Rio de Janeiro, Brazil

[5]Université de Strasbourg, CNRS UPR9022, Strasbourg, France

## AUTHOR ORCIDs

José Henrique M. Oliveira ⓘ http://orcid.org/0000-0003-3814-5312

## FUNDING

| Funder | Grant(s) | Author(s) |
| --- | --- | --- |
| Instituto Serrapilheira | 13452 | José Henrique M. Oliveira |
| Conselho Nacional de Desenvolvimento Científico e Tecnológico | 407312/2018, 403499/2021-6 | José Henrique M. Oliveira |

| Funder | Grant(s) | Author(s) |
|---|---|---|
| Fundação de Amparo à Pesquisa e Inovação do Estado de Santa Catarina | 2024TR002383, 2024TR002667 | José Henrique M. Oliveira |

## AUTHOR CONTRIBUTIONS

Mariana Maraschin, Formal analysis, Investigation, Writing – review and editing | Diego Novak, Data curation, Investigation | Valdorion José Klein Junior, Investigation | Lucilene W. Granella, Investigation | Luiza J. Hubner, Investigation | Athina R. Medeiros, Investigation | Tiago Gräf, Data curation | Guilherme Toledo-Silva, Data curation | Daniel S. Mansur, Formal analysis, Funding acquisition, Writing – review and editing | José Henrique M. Oliveira, Conceptualization, Data curation, Formal analysis, Funding acquisition, Supervision, Writing – original draft, Writing – review and editing

## ADDITIONAL FILES

The following material is available online.

### Supplemental Material

**Figure S1 (Spectrum00001-25-s0001.tiff).** DENV4 amino acid sequence.
**Figure S2 (Spectrum00001-25-s0002.tiff).** Maximum likelihood phylogenetic tree of DENV4 genotype 2.
**Figure S3 (Spectrum00001-25-s0003.tiff).** DENV4 viral loads offered to mosquitos and inside mosquitos immediately after feeding.
**Table S1 (Spectrum00001-25-s0004.tiff).** Amino acid differences between DENV4 strains used.

### Open Peer Review

**PEER REVIEW HISTORY (review-history.pdf).** An accounting of the reviewer comments and feedback.

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
