## [Reviewer comments · Microbiology Spectrum]

Microbiology Spectrum

A Laboratory-Adapted and a Clinical Isolate of Dengue Virus Serotype 4 Differently Impact *Aedes aegypti* Life-History Traits Relevant to Vectorial Capacity

Mariana Maraschin, Diego Novak, Valdorion José Klein Junior, Lucilene Granella, Luiza Hubner, Athina Medeiros, Tiago Graf, Guilherme Toledo-Silva, Daniel Mansur, and Jose Henrique Oliveira

Corresponding Author(s): Jose Henrique Oliveira, Universidade Federal de Santa Catarina

Review Timeline:

Submission Date:	January 1, 2025
Editorial Decision:	May 2, 2025
Revision Received:	June 10, 2025
Editorial Decision:	September 5, 2025
Revision Received:	September 13, 2025
Accepted:	September 28, 2025

Editor: Luciana Costa

Reviewer(s): Disclosure of reviewer identity is with reference to reviewer comments included in decision letter(s). The following individuals involved in review of your submission have agreed to reveal their identity: Wei Huang (Reviewer #2)

Transaction Report:

DOI: <https://doi.org/10.1128/spectrum.00001-25>

Re: Spectrum00001-25 (A Laboratory-Adapted and a Clinical Isolate of Dengue Virus Serotype 4 Differently Impact *Aedes aegypti* Life-History Traits Relevant to Vectorial Capacity)

Dear Dr. Jose Henrique M Oliveira:

Thank you for the privilege of reviewing your work. Below you will find my comments, instructions from the Spectrum editorial office, and the reviewer comments.

Dear Dr. Oliveira

Unfortunately, we had several difficulties securing enough reviewers to your manuscript. Thus, it was reviewed by a specialist in the field and me. Below you will find our observations, suggestions and concerns. The way the manuscript is presented does not meet the standards for publication in *Microbiology Spectrum*, thus major changes will be required for a new round of revision. Please respond to all comments/concerns suggested.

Editor observations:

During careful review of your manuscript several issues were raised which needs further clarification.

First, both the abstract and the introduction are too generic. Thus, the work's hypothesis did not become clear. The introduction never describes important points that could account for differences between the two viral isolates chosen by the authors to conduct the work. For instance, a summary of the differences in *in vitro* infectivity rates in vertebrate and mosquito cells should be presented. Also, which immunomodulatory capacities in vertebrate cells were observed and what were different? Do they trace back to any of the genomic differences between these two isolates? This is relevant information for the understanding of the author's hypothesis.

Based on the above, the following affirmative needs to be clarified. Why did authors hypothesize it? "We hypothesized that the two strains would differentially impact *Aedes aegypti* life-history traits under laboratory conditions."

Second, and the main concern regarding the manuscript, the first set of results presented regarding the viral genomic sequence characteristics and comparisons are based on previously published data from another group, not data generated by this work. No fundamental new analysis is presented. Therefore, relevant data to this paper should be summarized within introduction and /or rational leading to this manuscript results and referenced. Any additional relevant figure created by the authors should be presented as supplementary, making clear it is a re-analysis of previously published data.

Third, the rational for using the maximum available dose for each strain and a 1/10 dilution is not clear. The fact that the "clinical isolate" could not produce equivalent viral titers as the lab-adapted one and that limits the higher (desired) dose to be used is well accepted, however, a condition in which the same dose for both isolates is used needs be included. To evaluate author's hypothesis an equivalent dose and the highest possible dose of each makes more sense than the conditions chosen by the authors. Please, clarify this important point.

Still regarding the author's conclusion "Both strains were able to infect and replicate in *Aedes aegypti* (Figure 2B)." Considering that from 14 days after exposition viral loads should be increasing compared to the inoculum dose and achieve higher levels at 21 days, only around 15 to 30% of the mosquitos could maintain replication, except for the higher dose of the lab adapted isolate. Surprisingly, for the lowest dose of the clinical isolate at 14 days after infection 33% apparently sustained infection, however it was not maintained up to 21 days after exposition. Increase the n to confirm?

Considering what is expected, as referred by authors: "While DENV load increases 100-1000 fold in mosquitoes during its spread from the midgut to the salivary glands (Salazar et al. 2007)" the data presented does not allow the conclusion of sustained replication.

Thus, the affirmation that "these isolates replicate in mosquitoes independent of the inoculum" should also be reconsidered.

Were the following experiments analyzing mosquito survival, fertility, fecundity and flight activity done with the same infected groups presented in figure 2? If not, did authors check for the presence of viruses / viral loads in these mosquitos? It is especially relevant when analyzing the fertility experiment, since it shows a very wide variation in fertility for all groups but specifically for the clinical isolate one.

What is the relevance of measuring flight activity 21 days after exposition, since from figure 2, there wasn't the presence of DENV? This should be discussed in more details.

Discussion is too speculative, it is clear that the data regarding genomic characteristics of the clinical isolate is totally dissociated from the analysis/interpretation of the mosquito infection data.

Authors need to discuss study's limitations.

C6/36 and Vero cells origin need to be described.

Number of passages for the clinical isolate needs to be described.

Revision Guidelines

Sincerely,
Luciana Costa
Editor
Microbiology Spectrum

Reviewer #1 (Comments for the Author):

Comment 1. Overall structure in the abstract did not follow a particular systematic approach of the ideas the authors seek to convey. For example, line 34-37, How do the authors account for any potential loss in strain virulence in the lab-adapted strain of DENV and its impact on the tested parameters (fertility, flight, survival)?

Comment 2: Line 21, I suggest revising 'biotechnological' with 'biological approach' for clarity.

Comment 3: line 24, the word 'philogenetically' should be revised to 'phylogenetically'.

Comment 4. Line 44, 47, ...'Aedes aegypti' should be italicized and the entire text should be cross-checked to ensure consistency. Additionally, I suggest using recommended abbreviations for the genus name 'Aedes' once the full name has been introduced for the first time.

Comment 5. Line 79, delete 'In this manuscript'.

Comment 6. Line 84-85. I suggest including a reference to this statement.

Comment 7. Line 88. Revise 'mosquito health' to 'mosquito features such as fitness, survival,...'

Comment 8. Statement from Line 91-103 should be summarized into a single statement and may be elaborated upon in the discussion section.

Comment 9. Line 119, revise '...(collected in the presence of EDTA)...' to '...(collected into tubes containing EDTA(concentration?)...', and provide the concentration of EDTA in the collection tubes.

Comment 10. Line 131, I suggest revising 'Oven' to 'incubator'.

Comment. 11. Statement from Line 161-164 is difficult to follow and should be revised.

Comment 12. Line 173, Can the authors explain what was the other Vero cell type was and its optimization for DENV4 TVP/360?

Comment 13. Abbreviations used in the text for the first time need to be explained. What is 'DMEM, PFA, PBS, etc.?

Comment 14. Line 207, the sentence here is difficult to follow, what do the authors mean by '...whole blood, mock...?'

Comment 15. Authors should include a brief description of what a powdered dog chow is, the brand/producer and quantities used?

Comment 16. Line 277 and 301. To say "two doses" lacks clarity and does not ensure reproducibility of these findings. What was the viral titers administered?

Comment 17. Line 301-329. This paragraph is unnecessarily long, difficult to follow and reiterates the methodology/approach in what is supposed be a results section. This assertion holds true for the entirety of the results and discussion sections as well. I suggest a revision of these section.

Comment 18. Line 342-346. The study did not investigate mitochondrial respiration as an additional parameter to the mosquito characteristics being tested. However, the study draws analysis/conclusions based of this phenomenon stating 'reduction in mitochondrial respiration', which can be misleading.

Comment 19. Line 414-416. The statement here is misleading. Although mosquitoes could potentially transmit microbes, it is generally not conclusive to associate this ability with tolerance to the microbe as several extrinsic and intrinsic factors can contribute this capacity.

Comment 20. Line 419-422. Can the Authors explain exclusion of potential genetic factors playing a crucial role in a quantitative capacity to this phenomenon?

Comment 21. Line 428-444. These lengthy statements are not necessary, other than augmenting the word count. The same information has been repeated in the introduction section of the manuscript. Additionally, the entirety of the Discussion section focuses on previous reports to the neglect of the current study's own findings. I suggest a revision of this section.

Dear Dr. Luciana Costa. Thank you for the time to edit and review our manuscript. We appreciate the comments from you and reviewer 1 and believe they will improve the quality of the work. In the following lines, we provide a point-by-point explanation of all the changes we have made in this revised version.

Editor's comments:

1 - "First, both the abstract and the introduction are too generic. Thus, the work's hypothesis did not become clear. The introduction never describes important points that could account for differences between the two viral isolates chosen by the authors to conduct the work. For instance, a summary of the differences in *in vitro* infectivity rates in vertebrate and mosquito cells should be presented. Also, which immunomodulatory capacities in vertebrate cells were observed and what were different? Do they trace back to any of the genomic differences between these two isolates? This is relevant information for the understanding of the author's hypothesis. Based on the above, the following affirmative needs to be clarified. Why did authors hypothesize it? "We hypothesized that the two strains would differentially impact *Aedes aegypti* life-history traits under laboratory conditions."

We have extensively changed the abstract, importance, and introduction sections of the manuscript. All the changes are highlighted in the marked-up manuscript. We have added in the introduction a more detailed view of what is known about the differences regarding infectivity and immunity in cells (both insect and vertebrate). The section now reads:

"We evaluated the impact of two Dengue virus serotype 4 genotype II strains (DENV4) on different life-history traits of *Ae. aegypti*. The strains are a laboratory-adapted reference strain DENV4 - TVP/360 (GenBank - KU513442), originally isolated in the Dominican Republic in 1981 (Mackow et al. 1987; Durbin et al. 2001) and a clinical isolate from a patient presenting high serum viral load in southern Brazil, DENV4 - LRV 13/422 (GenBank - KU513441) (Sarathy et al. 2015; Kuczera et al. 2016). They were chosen because they exhibit different *in vitro* infectivity rates and immunomodulatory capacities in vertebrate cells (Kuczera et al., 2016). Specifically, the LRV 13/422 strain is able to infect C6/36 *Ae. albopictus* cells and Huh7.5 hepatocyte-derived human cells more efficiently than the TVP/360 strain. Also, LRV 13/422 is less sensitive to Interferon-alpha (IFN- α) inhibition in Huh7.5 cells and preliminary analyses revealed an extensive genetic divergence between the two strains (Kuczera et al., 2016). Lines 78 - 89.

We also tried to make it clear why we thought the differences observed in cells could reflect differences *in vivo*, in the mosquito vector. This section is now complemented by the explanations detailed above and now reads:

"Based on differences in *in vitro* infection efficiency, immune responses in vertebrate cells, and amino acid-level divergence between viral sequences, we hypothesized that the two strains would differentially affect *Aedes aegypti* life-history traits under laboratory conditions". Lines 90 - 92.

2 - Second, and the main concern regarding the manuscript, the first set of results presented regarding the viral genomic sequence characteristics and comparisons are based on previously published data from another group, not data generated by this work. No fundamental new analysis is presented. Therefore, relevant data to this paper should be summarized within introduction and /or rational leading to this manuscript results and referenced. Any additional relevant figure created by the authors should be presented as supplementary, making clear it is a re-analysis of previously published data.

For this work, we did not find it necessary to resequence the genomes of strains LRV13/422 and TVP/360, as we know and trust the work of the group of authors which generated it (Kuckzera et al., 2016). However, we believe that we could contribute with new and more detailed comparative phylogenetic analyses between the two strains, since it was superficially addressed by Kuckzera and only included as a supplementary table in their work. In agreement with your comment, we have acknowledged in the introduction that a previous work had showed evidence of genetic divergence between the two strains. We also moved Figure 1 and Table 1 to the supplementary material, now labelled as Supplementary Figure 1 and Supplementary Table 1, respectively. The Suppl. Figure 1 of the original version (phylogenetic tree) of the manuscript is now Supplementary Figure 2.

3 - Third, the rational for using the maximum available dose for each strain and a 1/10 dilution is not clear. The fact that the "clinical isolate" could not produce equivalent viral titers as the lab-adapted one and that limits the higher (desired) dose to be used is well accepted, however, a condition in which the same dose for both isolates is used needs be included. To evaluate

author's hypothesis an equivalent dose and the highest possible dose of each makes more sense than the conditions chosen by the authors. Please, clarify this important point.

Using the same infectious dose when comparing infection dynamics between groups is essential (Duneau et al., 2017 - <https://doi.org/10.7554/eLife.28298>), and our experimental design accounted for this. In this corrected version, we have included Supplementary Figure 3 to illustrate this point. There is no statistical difference in the viral titers in the blood meals provided to mosquitoes between the DENV4 TVP "low dose" and DENV4 LRV "high dose" groups, as shown by the analysis of independent biological replicates (Supplementary Figure 3A). Consistently, there is also no statistical difference in the amount of infectious virus ingested by *Aedes aegypti*, immediately after feeding (Supplementary Figure 3B). To make this point clear to the reader, we added the above explanation in the main text of the results section.

The segment now reads: "Using the same infectious dose when comparing infection dynamics between groups is essential, and our experimental design accounted for this. There is no statistical difference in the viral titers in the blood meals provided to mosquitoes between the DENV4 TVP "low dose" and DENV4 LRV "high dose" groups, as shown by the analysis of independent biological replicates (Supplementary Figure 3A). Consistently, there is also no statistical difference in the amount of infectious virus ingested by *Aedes aegypti* immediately after feeding (Supplementary Figure 3B)." Lines 281 – 288.

Regarding the rationale for using maximum available doses and a 1/10 dilution, it is known that virus titers in the blood offered to the mosquitoes strongly influence vector competence in ways that are not clear from a mechanistic standpoint (Hodoamedda et al., 2024 - <https://doi.org/10.1371/journal.ppat.1012047>). To characterize this phenomenon, we and others have used the dose variation strategy reported here to better describe *Aedes* vector competence (Maraschin et al., 2023 - DOI: 10.1016/j.jinsphys.2023.104573 and Novelo et al., 2019 - <https://doi.org/10.1371/journal.ppat.1008218>). We have stressed this in the results section to explain our experimental strategy. The segment now reads "Since the initial infectious dose strongly influences DENV infection in *Aedes aegypti* (Novelo et al., 2019), we used the maximum available dose for each strain and a 1/10 dilution. The infectious particle virus concentration (input in Figure 1C) is shown in red and expressed as plaque-forming units (PFU) per microliter of blood offered to mosquitoes (PFU/ul of blood)". (Lines 276 – 280).

Supplementary Figure 3

4 - Considering that from 14 days after exposition viral loads should be increasing compared to the inoculum dose and achieve higher levels at 21 days, only around 15 to 30% of the mosquitoes could maintain replication, except for the higher dose of the lab adapted isolate. Surprisingly, for the lowest dose of the clinical isolate at 14 days after infection 33% apparently sustained infection, however it was not maintained up to 21 days after exposition. Increase the n to confirm? Considering what is expected, as referred by authors: "While DENV load increases 100-1000 fold in mosquitoes during its spread from the midgut to the salivary glands (Salazar et al. 2007)" the data presented does not allow the conclusion of sustained replication. Thus, the affirmation that "these isolates replicate in mosquitoes independent of the inoculum" should also be reconsidered.

In the infection dynamics figure (Figure 1 of this revised version), "n.d." means "not determined". We haven't measured viral loads in mosquitoes fed the lowest dose of DENV4-LRV at 7 and 21 DPIs.

There is no example in the Aedes-arbovirus literature of mosquitoes that cleared a viral infection once they have been infected. If the mosquito is infected, it will be chronically infected throughout its lifetime. What changes, for reasons that are not clear in the literature and are not the scope here, is the fraction of the mosquito population that gets infected, referred to as the midgut infection barrier (Johnson et al., 2024 - <https://doi.org/10.1371/journal.ppat.1011975>). This is the reason why two complementary read outs are commonly used; infection intensity and

infection prevalence, as we did here. In both figures 1B and 1C we show, for both viral strains, at 14 days post-infection, we have viral replication (1C - intensity 14 DPI). We can also include the word “sustained” because by looking at the infection intensity (1C), once the mosquito gets infected, regardless of the initial dose, viral loads tend to be around 5×10^4 /mosquito for all the strains and doses tested. This is also illustrated in figure 3 of the beautiful manuscript Johnson et al., 2024 mentioned above.

Concerning the statement “*considering that from 14 days after exposition viral loads should be increasing compared to the inoculum dose*”, this is not always the case, as hosts have reported maximum carrying capacities to sustain viral replication, as observed in the DENV4/TVP higher dose kinetics (1B - dark green). The Salazar (2007) paper (DOI: 10.1186/1471-2180-7-9), used as an example of DENV infection dynamics in *Aedes aegypti* was done with the Chetumal strain of *Aedes aegypti*, known as a highly susceptible strain of *Aedes aegypti* (Bennett et al., 2002 - DOI: 10.4269/ajtmh.2002.67.85). The Salazar 2007 paper was mentioned because it is one of the few examples of DENV plaque assays performed in whole *Aedes aegypti* over a long percentage of the mosquito life span, as we did in our manuscript. To avoid generalizations, we have modified a sentence in the discussion section to say that virus loads increase 100-1000 in mosquitoes only in highly susceptible strains of *Aedes aegypti*. This segment now reads “While DENV load increases 100-1000 fold in susceptible mosquito strains during its spread from the midgut to the salivary glands (Salazar et al. 2007), *Ae. aegypti* does not experience significant fitness costs associated with chronic infection (Maraschin et al. 2023)...” (Lines 445 – 448).

5- Were the following experiments analyzing mosquito survival, fertility, fecundity and flight activity done with the same infected groups presented in figure 2? If not, did authors check for the presence of viruses / viral loads in these mosquitos? It is especially relevant when analyzing the fertility experiment, since it shows a very wide variation in fertility for all groups but specifically for the clinical isolate one.

It is technically impossible to perform survival, fertility, fecundity and flight activity assays with the same infected group of mosquitoes presented in figure 1 (figure 2 of the previous version of the manuscript). All the reported differences are group (population) effects. The midgut is the main infection barrier for arboviruses in mosquitoes. Researchers have two experimental possibilities. Feed mosquitoes with blood supplemented with high (and sometimes artificial)

doses of virus to guarantee the highest % of infection possible (rarely 100%) or bypass the midgut barrier with a systemic virus injection (Olmo, 2018 - <https://doi.org/10.1038/s41564-018-0268-6>). These are the standard approaches in the mosquito-infection biology protocols. Since our group is interested in the influence of blood components during infection, we use the physiological feeding route of infection. This approach mimics the natural entry of the virus and its immune response. It has the caveat of not leading to 100% mosquito infectivity, leading us and others to analyze group effects and mean/median readouts. There will always be some heterogeneity in the replicates that make up an experimental group. Despite this, we observe several virus-induced alterations in the readouts we measured, such as fecundity, fertility, and flight capacity. All of them represent population effects.

6 - What is the relevance of measuring flight activity 21 days after exposition, since from figure 2, there wasn't the presence of DENV? This should be discussed in more details.

As mentioned in commentary 4, we haven't measured viral loads in mosquitoes at 21 DPI, but since we observe a 35% infection prevalence, with an average of 5×10^4 PFU/mosquito in the infected mosquitoes of the DENV/LRV - low dose exposed group, we believe our INFLATE result at 21DPI is relevant to demonstrate that we measured strain-specific effects in the mosquito induced flight capacity in the group exposed/infected by the TVP strain, but not the LRV strain, in a time point highly relevant to vector capacity.

7 - Discussion is too speculative, it is clear that the data regarding genomic characteristics of the clinical isolate is totally dissociated from the analysis/interpretation of the mosquito infection data. Authors need to discuss study's limitations.

We restructured and reduced the discussion section. We removed the section discussing our results in the context of the deposited viral sequences of TVP and LRV strains. We also removed the entire 4th paragraph of the original text (“The ability of dengue virus to infect...”) where we tried to speculate on the strain-specific differences in infection prevalence. We have added a segment to discuss the limitations of our study (Lines 464 – 470).

The last paragraph of the discussion section now reads:

“Some of the limitations of our study include the fact that we have demonstrated viral strain-induced changes in parameters associated with *Ae. aegypti* vector competence and physiology under laboratory conditions, and not in field settings with the influence of real-world ecological parameters. Also, feeding mosquitoes with decreasing doses of arboviruses leads to reduced percentages of infections (Figure 1C). The observed differences should be interpreted with caution and understood as group or population effects”.

8 - C6/36 and Vero cells origin need to be described. Number of passages for the clinical isolate needs to be described.

Both information were added.

The origin of the cells used was added to sections 4.2 and 4.5 of Materials and Methods.

Vero cells (ATCC, CCL-81™).

Vero E6 cells (ATCC, CRL-1586).

C6/36 cells (ATCC, CRL-1660).

Concerning the number of passages of the clinical isolate: the DENV4 LRV13/422 strain used was at passage 5. This information was added to section 4.2 of the Materials and Methods.

Reviewer #1

Comment 1. Overall structure in the abstract did not follow a particular systematic approach of the ideas the authors seek to convey. For example, line 34-37. How do the authors account for any potential loss in strain virulence in the lab-adapted strain of DENV and its impact on the tested parameters (fertility, flight, survival)?

Please see the response to commentary 1 of the editor. We have extensively modified the abstract in an attempt to improve the flow connecting the use of different viral strains and their consequences for the mosquito host. We acknowledge that prolonged laboratory passage can potentially lead to changes in viral characteristics, including virulence. While we utilized a lab-adapted strain of DENV4, which may exhibit altered virulence, our study focuses on

characterizing the interaction between a specific virus serotype and the mosquito vector under controlled laboratory conditions to elucidate fundamental aspects of their interaction that are relevant to vector competence.

Comment 2: Line 21, I suggest revising 'biotechnological' with 'biological approach' for clarity.

Changed.

Comment 3: line 24, the word 'philogenetically' should be revised to 'phylogenetically'.

Changed.

Comment 4. Line 44, 47, ...'Aedes aegypti' should be italicized and the entire text should be cross-checked to ensure consistency. Additionally, I suggest using recommended abbreviations for the genus name 'Aedes' once the full name has been introduced for the first time.

Changed.

Comment 5. Line 79, delete 'In this manuscript'.

Deleted.

Comment 6. Line 84-85. I suggest including a reference to this statement.

We have added the reference. Kuczera et al. 2016.

Comment 7. Line 88. Revise 'mosquito health' to 'mosquito features such as fitness, survival,

We have removed the word “health”.

Comment 8. Statement from Line 91-103 should be summarized into a single statement and may be elaborated upon in the discussion section.

We summarized the sentence, which now reads: We found strain-specific phenotypes in the infection prevalence, fertility, and INFLATE at certain time points following infection (Lines 97 – 98). We shortened the last sentence of the introduction section that now reads: “To modulate mosquito disease tolerance, we need to improve our ability to quantify mosquito fitness

and homeostatic status during infection, and to understand the molecular basis of strain-specific effects on vector life-history traits". Lines 99 – 102.

Comment 9. Line 119, revise '...(collected in the presence of EDTA)...' to '...(collected into tubes containing EDTA(concentration?)...'; and provide the concentration of EDTA in the collection tubes.

Corrected. The segment now reads "collected into 5 mL tubes containing 7mg of K₂EDTA".

Comment 10. Line 131, I suggest revising 'Oven' to 'incubator'.

Changed.

Comment. 11. Statement from Line 161-164 is difficult to follow and should be revised.

Under laboratory conditions, *Ae aegypti* is kept with *ad libitum* access to 10% sucrose solutions for energy demands and hydration. To blood feed the mosquitoes, 18-24 hours prior to the blood meal, the females are starved to increase the blood feeding success percentage. We observed that when we simply remove the sucrose solution, some females might experience dehydration stress before the meal, also reducing the % of blood-fed females. By replacing the sucrose solution to water, we allow the females to starve for 18-24 hours but do not expose the mosquitoes to water deprivation, only carbohydrate deprivation. The sentence in the text now reads: "*Ae. aegypti* females were fasted for 18-24 hours before a blood meal. During this fasting period, mosquitoes had access to water (but not sucrose solution) before being offered a meal containing a 1:1 mixture of RBC cells and L-15 medium with either DENV4 TVP/360 virus or DENV4 LRV13/422" (Lines 161 – 165).

Comment 12. Line 173, Can the authors explain what was the other Vero cell type was and its optimization for DENV4 TVP/360?

We have added to section 4.5 the details of each cell type used. The section now reads "Vero cells (ATCC, CCL-81) were used to quantify DENV4 TVP/360 and Vero E6 cells (ATCC, CRL-1586) were used to quantify DENV4 LRV 13/422 based on plaque optimization for each viral strain". Vero E6 is a commercially available clone (clone E6) of the Vero ATCC C1008 (<https://www.atcc.org/products/crl-1586>). As it is known, plaque assays for DENV can be tricky and prone to inconsistencies, variations among serum lots, manufacturers, etc. We observed

that the consistency, replicability, and resolution of plaques (after crystal violet staining) of our DENV4 LRV 13/422-derived samples were higher when using clone E6 of the Vero ATCC C1008.

Comment 13. Abbreviations used in the text for the first time need to be explained. What is 'DMEM, PFA, PBS, etc.?

Corrected.

Comment 14. Line 207, the sentence here is difficult to follow, what do the authors mean by '...whole blood, mock...?

Those definitions were present in the materials and methods section, item 4.4. To improve clarity, under the materials and methods item 4.7 (*Ae. aegypti* fecundity and fertility), we refer to item 4.4, where we define the feeding conditions tested in the fecundity/fertility group of experiments.

Comment 15. Authors should include a brief description of what a powdered dog chow is, the brand/producer and quantities used?

We added the information requested by the reviewer. The segment now reads: Larvae were maintained in filtered, dechlorinated tap water at a density of approximately 100 larvae per liter. They were fed Purina™ powdered dog chow, with a total of 500 mg provided in 2–3 additions over the course of the larval stage, without changing the water. Lines 109 – 112.

Comment 16. Line 277 and 301. To say "two doses" lacks clarity and does not ensure reproducibility of these findings. What was the viral titers administered?

Please, see our response above to the commentary 3 of the editor. This segment was changed to clearly state the administered doses. A new supplementary figure (SP3) was added. Also, in Figure 1C of this revised version, all the input doses offered to the mosquitoes are presented "Input - infectious doses (PFU/ul of blood)". Also, in Figure 1B (of this revised version), the viral infectious particles ingested by each mosquito are presented for all the strains and doses as PFU/mosquito at T0. The text now reads: "Using the same infectious dose when comparing infection dynamics between groups is essential, and our experimental design accounted for this. There is no statistical difference in the viral titers in the blood meals provided to mosquitoes

between the DENV4 TVP "low dose" and DENV4 LRV "high dose" groups, as shown by the analysis of independent biological replicates (Supplementary Figure 3A). Consistently, there is also no statistical difference in the amount of infectious virus ingested by *Aedes aegypti* immediately after feeding (Supplementary Figure 3B)." Lines 280 – 287.

Comment 17. Line 301-329. This paragraph is unnecessarily long, difficult to follow and reiterates the methodology/approach in what is supposed be a results section. This assertion holds true for the entirety of the results and discussion sections as well. I suggest a revision of these section.

We restructured the section, reducing its overlapping content with the materials and methods. The section now reads: "The impact of virus infection on parameters directly connected to vector fitness was evaluated following mosquito challenge with both DENV4 strains. Feeding *Ae. aegypti* with blood supplemented with different concentrations of infectious particles did not alter the total number of eggs laid by each fully engorged female (Figure 3A). In Figure 3, "blood" represents whole human blood (RBCs + plasma), which has a higher total protein concentration than mock, as evidenced by a greater fecundity of *Ae. aegypti* (blood ~70 eggs/female vs mock ~35 eggs/female). Challenging mosquitoes with the clinical strain of DENV4 (LRV) resulted in a significant reduction in the percentage of eclosion (~50%), irrespective of the dose tested (Figure 3B). This result suggests a lower adaptation of the LRV strain, which is consistent with its recent interaction with our colony mosquitoes (Red Eyes strain) instead of field mosquitoes circulating in southern Brazil (de Oliveira et al. 2023). On the other hand, we did not observe statistically significant differences in fertility with the laboratory-adapted TVP strain, consistent with the higher adaptability of this virus under laboratory conditions. Interestingly, there is no difference in fertility between blood-fed and mock-fed mosquitoes, averaging around 70% eclosion, suggesting that females can optimize fecundity (egg production) according to the nutritional status of the meal to maximize fertility, similar to what has been described for desiccation stress in *Ae. aegypti* (Prasad et al. 2023)." Lines 310 – 327.

Comment 18. Line 342-346. The study did not investigate mitochondrial respiration as an additional parameter to the mosquito characteristics being tested. However, the study draws

analysis/conclusions based of this phenomenon stating 'reduction in mitochondrial respiration', which can be misleading.

The reported study, Gonçalves 2009, was performed in isolated *Ae aegypti* flight muscle mitochondria and demonstrated a reduction in mitochondrial function 24 hours after a blood meal. We observed a phenotype directly connected to a reduction in the induced-flight capacity in uninfected 24 hours after feeding (when the midgut is full of blood) (Figure 4B of this revised version) when compared to uninfected mosquitoes 7 days after feeding (when the midgut has no blood content since digestion has finished) (Figure 4C of this revised version). We reasoned that this reduction in flight is consistent with the results demonstrated by Gonçalves 2009. We did not conclude that the observed reduction in INFLATE 24 hours after uninfected blood ingestion is due to reduced mitochondrial respiration. Following the reviewer's suggestion, we removed the sentence "Consistent with a reduction in mitochondrial respiration following a blood meal (Gonçalves et al. 2009)".

Comment 19. Line 414-416. The statement here is misleading. Although mosquitoes could potentially transmit microbes, it is generally not conclusive to associate this ability with tolerance to the microbe as several extrinsic and intrinsic factors can contribute this capacity.

We have restructured the sentence. The segment now reads: "Mosquitoes can harbor and transmit microbes such as dengue virus and Plasmodium spp. without exhibiting overt signs of pathology, a phenomenon often described as disease tolerance. While this trait has been relatively well characterized phenotypically, its underlying molecular mechanisms remain poorly understood". Lines 403 – 406.

Comment 20. Line 419-422. Can the Authors explain exclusion of potential genetic factors playing a crucial role in a quantitative capacity to this phenomenon?

Genetic factors will influence disease tolerance. We state that there is a defined method to measure disease tolerance. This method is the reaction norm of a dose-response curve correlating fitness and microbe load. The slope of this curve is tolerance. This has been proposed by Ellen L. Simms in a seminal article extensively used in the field of ecological immunology and evolutionary ecology. The focus here is on quantifying tolerance rather than understanding its underlying mechanisms.

Simms, 2000 - <https://doi.org/10.1023/A:1010956716539>.

Comment 21. Line 428-444. These lengthy statements are not necessary, other than augmenting the word count. The same information has been repeated in the introduction section of the manuscript. Additionally, the entirety of the Discussion section focuses on previous reports to the neglect of the current study's own findings. I suggest a revision of this section.

We disagree with the reviewer's comment. The statements in question are essential to place our findings in the appropriate scientific context. DENV4 remains poorly characterized in mosquitoes and has been reported to displace DENV1 in mosquito populations, underscoring the relevance of studies like ours. Furthermore, DENV4 has been detected in mosquitoes in areas without reported human cases, which highlights the need to investigate mosquito–virus interactions specifically involving DENV4 and *Aedes aegypti*, as we have done here. We do not believe these statements were added merely to increase word count. On the contrary, they were carefully included based on a thorough review of the literature and the work time of the researchers involved in the manuscript to help readers understand the significance of our findings.

We have reduced the discussion section by cutting the entire segment from lines 462-496 of the original pdf file.

Re: Spectrum00001-25R1 (A Laboratory-Adapted and a Clinical Isolate of Dengue Virus Serotype 4 Differently Impact *Aedes aegypti* Life-History Traits Relevant to Vectorial Capacity)

Dear Dr. Jose Henrique M Oliveira:

Thank you for the privilege of reviewing your work. Below you will find my comments, instructions from the Spectrum editorial office, and the reviewer comments.

I apologize for the delay in analyzing the reviewed version of the manuscript "Laboratory-Adapted and a Clinical Isolate of Dengue Virus Serotype 4 Differently Impact *Aedes aegypti* Life-History Traits Relevant to Vectorial Capacity" in part due to the difficulty in securing reviewers for the second round of revision.

The authors presented a much-improved manuscript version, following through on most of the reviewers' suggestions and comments. This is a relevant piece of data, and thus, I still feel that authors can be more precise when describing results in Figure 1 (lines 345 - 349) and the following conclusion /discussion of it (lines 574-580 of the discussion):

"We were interested in two readouts, the infection prevalence, defined as the percentage of infected mosquitoes, and infection intensity, defined as the amount of infectious particles per mosquito. Figure 12C shows that despite a difference in input dose varying by a factor of 100X (from TVP - higher dose to LRV - lower dose), infection intensity had a minor impact on virus titer per mosquito."

Authors should tone down the "minor impact" since the study presents no evidence that this is the case. Moreover, the difference reached statistical significance. Thus, having an impact or not cannot be ruled out either way.

"In Figure 12C we show that despite a marked difference in prevalence at 14DPI between the LRV 13/422 and TVP/360 strain depending on the initial DENV4 input dose, infection intensity (Figure 12C - middle panel) was not influenced. This result suggests that immune resistance plays a minor role in the DENV4 replication cycle within the mosquito and unknown factors controlling the midgut infection barrier might be more important to mosquito infection and, therefore, disease transmission." Again, for the highest input dose there was a statistical significance in the intensity readout, thus authors cannot affirm that "intensity was not influenced".

For the manuscript to be accepted, authors need to review these incongruencies in data description, limit the conclusions to what is supported by the data and then discuss from their point of view and from what is already in the literature that the possible impact on vector viability could not be directly influenced by virus input, but also influenced by other characteristics of the vector infection.

A discussion on the influence of virus adaptation in cell culture should also be contemplated, since these differences in the dynamics of vector infection for the adapted viral isolate could be an exclusive event not expected to occur naturally.

Revision Guidelines

Data availability: ASM policy requires that data be available to the public upon online posting of the article, so please verify all

links to sequence records, if present, and make sure that each number retrieves the full record of the data. If a new accession number is not linked or a link is broken, provide Spectrum production staff with the correct URL for the record. If the accession numbers for new data are not publicly accessible before the expected online posting of the article, publication may be delayed; please contact production staff (Spectrum@asmusa.org) immediately with the expected release date.

Sincerely,
Luciana Costa
Editor
Microbiology Spectrum

Reviewer #2 (Comments for the Author):

This manuscript assesses effects on life history traits (mortality, fecundity and flight capacity) and vector competence-associated parameters of two DENV4 strains: the laboratory-adapted DENV4-TVP/360 strain and clinical isolate DENV4-LRV13/422. While addressing a knowledge gap for DENV4, its most critical flaw is that it does not provide evidence of connections to actual transmission potential to translate into real-world public health relevance.

Major Weaknesses and Issues

1. Weak Molecular Mechanism Research, Unclear Causes of Phenotypic Differences

Specific phenotypes emerged (the TVP strain increases flight ability at 1 dpi and 21 dpi; the LRV strain decreases fertility), but not their underlying molecular pathways.

2. Lack of Background Information on the Clinical Strain, Uncertain Relevance to Natural Infections

The background introduction of the clinical strain DENV4-LRV13/422 is too brief. It only stated that this virus strain "was isolated from a patient with high serum virus load in southern Brazil, 2013", but no important information such as whether the condition of the patient was serious or not, and if this virus strain had been transmitted by mosquitoes locally and circulated concurrently with other three serotypes; so we are hard to find some connection between "clinically prevalent strains - mosquito transmission efficiency - human infection risk," thus offering little guidance for the practice of public.

3. Limited Vector Competence Validation

Viral vector competence is based only on infection prevalence and intensity (whole mosquito virus loads), not through salivary gland invasion and shedding. E.g., the manuscript does not evaluate whether better flight performance of DENV4-TVP/360 correlates with enhanced rates of salivary gland infections when mosquitoes would be infected at 21 dpi; rather, those correlations were only a phenotype (flight capacity) vs transmission outcomes.

Dear Dr. Luciana Costa,

Thank you for handling the manuscript once more. We are glad that you found that the reviewed version of the manuscript presents a clearer picture. We are confident that with the modifications in the current version, we will meet the publication standards of Microbiology Spectrum.

Editor's comments:

1 - The authors presented a much-improved manuscript version, following through on most of the reviewers' suggestions and comments. This is a relevant piece of data, and thus, I still feel that authors can be more precise when describing results in Figure 1 (lines 345 - 349) and the following conclusion /discussion of it (lines 574-580 of the discussion):

"We were interested in two readouts, the infection prevalence, defined as the percentage of infected mosquitoes, and infection intensity, defined as the amount of infectious particles per mosquito. Figure 12C shows that despite a difference in input dose varying by a factor of 100X (from TVP - higher dose to LRV - lower dose), infection intensity had a minor impact on virus titer per mosquito.

"Authors should tone down the "minor impact" since the study presents no evidence that this is the case. Moreover, the difference reached statistical significance. Thus, having an impact or not cannot be ruled out either way."

"In Figure 12C we show that despite a marked difference in prevalence at 14DPI between the LRV 13/422 and TVP/360 strain depending on the initial DENV4 input dose, infection intensity (Figure 12C - middle panel) was not influenced. This result suggests that immune resistance plays a minor role in the DENV4 replication cycle within the mosquito and unknown factors controlling the midgut infection barrier might be more important to mosquito infection and, therefore, disease transmission."

Again, for the highest input dose there was a statistical significance in the intensity readout, thus authors cannot affirm that "intensity was not influenced".

For the manuscript to be accepted, authors need to review these incongruencies in data description, limit the conclusions to what is supported by the data and then discuss from their point of view and from what is already in the literature that the possible impact on vector viability could not be directly influenced by virus input, but also influenced by other characteristics of the vector infection. A discussion on the influence of virus adaptation in cell culture should also be contemplated, since these differences in the dynamics of vector infection for the adapted viral isolate could be an exclusive event not expected to occur naturally.

Response to editor's comments:

As requested by the editor, we have modified both segments to provide a more precise description of our data. The first segment now reads:

We measured two readouts: infection prevalence, the percentage of infected mosquitoes, and infection intensity, the amount of infectious particles per mosquito. Overall, infection intensity showed no dose-dependent increase with input dose in all the experimental conditions. For the DENV4 TVP strain, mosquitoes ingested either 2.2×10^5 PFUs ($\pm 1.3 \times 10^4$ S.E.M., $n = 30$) at the highest dose or 4.8×10^3 PFUs ($\pm 8.3 \times 10^2$ S.E.M., $n = 28$) at the lowest dose. Despite this 46,000-fold difference in input, infection intensity at 14 DPI was not significantly different ($p = 0.326$, K–S test). The same pattern was observed with the DENV4 LRV strain. At the highest dose, mosquitoes ingested 3.6×10^3 PFUs ($\pm 6.0 \times 10^2$ S.E.M., $n = 40$), compared to 5.4×10^2 PFUs ($\pm 2.5 \times 10^2$ S.E.M., $n = 15$) at the lowest dose. Here too, despite a 6.6-fold increase in virus input, infection intensity at 14 DPI did not differ ($p > 0.999$, K–S test). Taken together, input doses ranging across $\sim 4 \times 10^5$ PFUs (a 407,781-fold difference between the TVP high dose and the LRV low dose) had little effect on virus titer per mosquito at 14 DPI. The only significant difference was detected at 14 DPI when comparing the highest doses of TVP and LRV.

Lines 289 - 302 of the corrected version.

Regarding the discussion section related to Figure 1C, now the segment reads “In Figure 1C we show that despite a marked difference in prevalence at 14DPI between the LRV 13/422 and TVP/360 strain, depending on the initial DENV4 input dose, infection intensity (Figure 1C - middle panel) exhibited a modest dose-dependency.

Lines 460 - 463 of the corrected version.

Regarding the limitations of the study, we added a sentence related to the influence of virus adaptation in cell culture in the last paragraph of the discussion section. It reads “The DENV4 TVP/360 strain is laboratory-adapted and reaches high titres in cell culture. It remains possible that the observed infection prevalence (Figure 1C) and induced-flight activity (Figure 4) result from artificial adaptation in cell culture”.

Lines 474 - 477 of the corrected version.

Response to Reviewer #2 comments:

1. Weak Molecular Mechanism Research, Unclear Causes of Phenotypic Differences. Specific phenotypes emerged (the TVP strain increases flight ability at 1 dpi and 21 dpi; the LRV strain decreases fertility), but not their underlying molecular pathways.

We recognize our work does not investigate the molecular causes of the observed phenotypes and acknowledge the importance of such experiments to a broader understanding of the Aedes vector competence. While this is beyond the scope of the current manuscript, studies are underway to address this limitation.

2. Lack of Background Information on the Clinical Strain, Uncertain Relevance to Natural Infections. The background introduction of the clinical strain DENV4-LRV13/422 is too brief. It only stated that this virus strain "was isolated from a patient with high serum virus load in southern Brazil, 2013", but no important information such as whether the condition of the patient was serious or not, and if this virus strain had been transmitted by mosquitoes locally and circulated concurrently with other three serotypes; so we are hard to find some connection between "clinically prevalent strains - mosquito transmission efficiency - human infection risk," thus offering little guidance for the practice of public.

Our work aimed to characterize life history traits of *Aedes aegypti* infected with two strains of an underexplored DENV serotype in vector biology. The manuscript Kuczera et al., 2016 (doi.org/10.1186/s12985-016-0548-9) isolated the DENV4 - LRV13/422 clinical strain and compared it in a mammalian cell culture system to a laboratory-adapted DENV4 strain frequently used (TVP/360). All the information we had from the original patient was obtained from Kuczera (2016). As the present manuscript focuses on vector life-history traits, we believe that the lack of clinical details about the patient, as well as the ecological relationship of the LRV13/422 strain with other co-circulating arboviruses, does not compromise the main conclusions of this study.

3.Limited Vector Competence Validation. Viral vector competence is based only on infection prevalence and intensity (whole mosquito virus loads), not through salivary gland invasion and shedding. E.g., the manuscript does not evaluate whether better flight performance of DENV4-TVP/360 correlates with enhanced rates of salivary gland infections when mosquitoes would be infected at 21 dpi; rather, those correlations were only a phenotype (flight capacity) vs transmission outcomes.

We appreciate the reviewer's comment and recognize that the suggested experiments would enrich the manuscript. However, measuring viral loads through plaque assays in whole-body mosquitoes can be used as an initial assessment of vector competence (Salazar et al., 2007 – doi:10.1186/1471-2180-7-9). In fact, the majority of articles describing arbovirus vector

competence dynamics (Novelo et al., 2019 – <https://doi.org/10.1371/journal.ppat.1008218>) and mechanisms only measure viral loads using qPCR. It is also recognized in the vector biology literature that there is no standard way to assess vector competence (Wu et al., 2022 – <https://doi.org/10.1038/s41597-022-01741-4>). This can be seen as a two-way route: on the one hand, it slightly limits comparisons between studies, but on the other hand, it broadens the field, preventing valuable data from being disregarded just because they do not fit a single standardized approach.

Re: Spectrum00001-25R2 (A Laboratory-Adapted and a Clinical Isolate of Dengue Virus Serotype 4 Differently Impact Aedes aegypti Life-History Traits Relevant to Vectorial Capacity)

Dear Dr. Jose Henrique M Oliveira:

Your manuscript has been accepted, and I am forwarding it to the ASM production staff for publication. Your paper will first be checked to make sure all elements meet the technical requirements. ASM staff will contact you if anything needs to be revised before copyediting and production can begin. Otherwise, you will be notified when your proofs are ready to be viewed.

Sincerely,
Luciana Costa
Editor
Microbiology Spectrum